# Development of a Mobile Laser Measurement System for Subway Tunnel Deformation Detection

**DOI:** 10.3390/s25020356

**Published:** 2025-01-09

**Authors:** Wei Li, Haoran Duan, Qiuzhao Zhang, Jiahui Liang, Wei Duan, Kaikun Zhang, Wenda Wang, Huachao Yang

**Affiliations:** 1School of Environment and Spatial Informatics, China University of Mining and Technology, Xuzhou 221000, China; liwei23@cumt.edu.cn (W.L.); ts21160116p31@cumt.edu.cn (H.D.); qiuzhao.zhang@cumt.edu.cn (Q.Z.); ts21160150p31@cumt.edu.cn (J.L.); lb21160002@cumt.edu.cn (W.D.); lb21160010@cumt.edu.cn (K.Z.); wwd@cumt.edu.cn (W.W.); 2Nanjing Institute Surveying, Mapping & Geotechnical Investigation, Co., Ltd., Nanjing 210019, China

**Keywords:** mobile laser measurement, tunnel detection, point cloud data processing

## Abstract

In recent years, mobile laser measurement systems have markedly enhanced the capabilities of deformation detection and defect identification within metro tunnels, attributed to their superior efficiency, precision, and versatility. Nevertheless, challenges persist, including substantial equipment costs, inadequate after-sales support, technological barriers, and limitations in customization. This paper develops a mobile laser measurement system that has been specifically developed for the purpose of detecting deformation in metro tunnels. The system integrates multiple modules, comprising a rail inspection vehicle, a three-dimensional laser scanner, an odometer, and an inclinometer, to facilitate multi-sensor temporal synchronization. By leveraging data from the inclinometer and odometer, the system performs point cloud coordinate corrections and three-dimensional linear reconstructions. Experiments conducted on the Xuzhou Metro validate the reliability and stability of the system, demonstrating its capability to meet routine deformation detection requirements. To improve deformation detection utilizing point cloud data, a pre-processing algorithm has been proposed, which incorporates point cloud denoising, centerline calculation, roadbed removal, and relative positioning based on mileage. Disturbed points are systematically identified and eliminated, while the convergence of tunnel sections and inter-ring misalignment are evaluated through ellipse fitting. Furthermore, to address encroachments upon tunnel locomotive limits, encroachment points and associated information are identified using the ray method. In conclusion, the proposed mobile laser measurement system offers an efficient and reliable solution for metro tunnel deformation detection, with significant potential for broader applications and future advancements.

## 1. Introduction

As of December 2023, a total of 55 cities in China have inaugurated and currently operate 306 urban rail transit lines, encompassing an extensive operational length of 10,165.7 km, with metro systems constituting over 75% of this total, thereby making it the largest network globally [1]. Owing to the cumulative effects of human activities, environmental factors, and other influences, metro tunnels may undergo alterations in shape, size, or spatial position, resulting in a variety of issues, including segment misalignment, lining cracks and damages, segment corrosion, lining leakage, longitudinal settlement deformation, and transverse convergence deformation [2,3], all of which pose significant threats to the safety of metro operations. The health condition of metro tunnels progressively deteriorates over time during their operational service. Consequently, performing regular structural condition assessments of shield tunnel facilities represents the most effective approach to accurately reflect the current state of the tunnels.

Presently, the most widely employed methods for detecting structural deformation and defects in metro tunnels encompass traditional geodetic techniques, sensor methodologies, and three-dimensional laser scanning technologies. Traditional methods for detecting tunnel deformation and defects utilize instruments such as total stations and levels for both manual and automated measurements [4,5,6]. This approach incurs substantial costs, lacks three-dimensional measurement outcomes, and fails to satisfy the requirements for high efficiency, precision, digitalization, and cost-effective inspections within extensive networks. The sensor methodology employs distributed fiber optic strain sensors, wireless inclination sensors, thermal infrared cameras, CCD cameras, and digital cameras for the detection of structural deformation and defects in tunnels. However, this method is comparatively constrained and necessitates stringent environmental conditions [7,8,9]. In contrast, three-dimensional laser scanning technology mitigates the operational risks inherent to traditional measurement methods and sustains stability under challenging external conditions, such as low light, complex environments, or multiple light sources [10,11,12,13,14]. In practice, this technology can be categorized into stationary and mobile scanning modalities. Stationary laser scanning technology necessitates frequent setup modifications, leading to delayed data acquisition and substantial data volumes, thereby limiting efficiency for routine metro inspections. Recently, mobile laser measurement technology has experienced rapid advancements in the field of tunnel inspection. Typically, such systems comprise laser scanners, odometers, inclination sensors, inertial navigation systems, and other components, achieving scanning rates of up to millions of points per second for comprehensive coverage of tunnel inner walls. Notable examples include the Leica Sitrack-one system developed by Leica, the Amberg GRP system by Swiss Amberg, the TLSD system co-developed by Tongji University and Capital Normal University, and the MS100 system produced by Southern Surveying and Mapping Company. Currently, mobile laser measurement systems are extensively utilized for routine inspections of operational metro tunnels in cities such as Beijing, Shanghai, and Nanjing. However, challenges persist, including substantial equipment costs, insufficient after-sales support, technological limitations, and a dearth of options for secondary customization.

## 2. Materials and Methods

### 2.1. Development and Application of a Mobile Laser Measurement System for Metro Tunnels

This paper details the development of a mobile laser measurement system for metro tunnels, comprising a track inspection vehicle, a control system, a power supply unit, a three-dimensional laser scanner, an odometer, and inclination sensors. The design encompasses the integration of the overall mechanical structure of the modules along with the communication scheme. The control system synchronizes the sensors by utilizing pulsed signals emitted from the scanner to achieve precise time synchronization. By employing the synchronized data from the odometer and inclination sensors, the original two-dimensional cross-sectional point cloud data are “stretched” and corrected for coordinates, ultimately resulting in high-precision three-dimensional point cloud data of the tunnel walls.

#### 2.1.1. Hardware Integration of Mobile Laser Measurement Systems

The mobile laser measurement system presented in this paper employs the FARO S150 3D laser scanner, which boasts a maximum acquisition rate of 976,000 points per second, a ranging accuracy of 1 mm, and a maximum vertical scanning speed of 97 Hz. The two-dimensional inclination sensors exhibit a measurement range of ±10°, an absolute accuracy of 0.001°, and a maximum measurement frequency of 100 Hz. Angle encoders are integrated with a microcontroller unit (MCU) to facilitate the construction of the odometer. The vehicle body is designed with a lightweight concept, while the scanner is mounted on an independent fixed bracket to facilitate ease of disassembly and installation. The inclination sensor, battery system, and control system are all securely housed within the enclosure, with the centers of the scanner, inclinometer, and vehicle body meticulously aligned along the same axis. To accommodate outdoor rail transit applications, two mounting positions for GNSS antennas have been designed above the vehicle body.

Figure 1 shows the physical model of the mobile laser measurement system. The system uses a wireless remote control to communicate with the receiver on the vehicle body, enabling forward, backward, and stop commands. Additional buttons on the vehicle provide functions such as emergency stop, mileage counting and reset, and controlling the lights.

The vehicle weighs about 50 kg and operates at constant speeds of 0.3 m/s, 0.6 m/s, or 1 m/s, running continuously for up to 4 h and collecting around 14 km of tunnel point cloud data per session. It requires only 2–3 operators. The system is controlled by an STM32F1X-based controller (Figure 2), which manages data collection, storage, and communication with the inclination sensor and odometer via a serial port. The controller integrates collected data, transmits the data to a host computer for real-time storage, and provides backup with its built-in storage. It sends pulse signals to the 3D laser scanner to control lens rotation and data recording and receives synchronization pulses for real-time calibration. Additionally, the controller monitors the circuit system and displays battery status, speed, and travel distance on a touch screen.

#### 2.1.2. Time Synchronization of Mobile Laser Measurement Systems

The laser scanner in this paper supports both CAN and TTL communication in the mobile scanning mode. CAN communication offers high reliability and flexibility, allowing external circuits to transmit pre-triggered CAN messages with Trigger_In signals at customizable frequencies. This method can theoretically assign an external time reference to each point cloud data point. However, achieving maximum data density and accuracy requires the system to operate at the highest sampling frequency. The scanner used in this paper can acquire up to 976,000 points per second, meaning CAN mode would need to process 976,000 CAN messages containing trigger signals and clock information every second. This demands the controller chip to handle each message within about 1 microsecond, creating an overwhelming processing challenge. By comparison, TTL mode uses fixed synchronization pulses from the scanner for external sensor synchronization. The laser head rotates around 100 times per second, resulting in a time interval of approximately 10 milliseconds between synchronization signals. This significantly lowers processor demands and reduces the risk of data loss, making TTL mode a more practical option for high-frequency operations.

Considering the advantages and disadvantages of both CAN and TTL modes, this paper adopts TTL mode for time synchronization between multiple sensors. Prior to operation, the external circuit system initially configures the scanner and inclinometer in accordance with the communication protocol and transmits two low-level pulse signals to the scanner via the Trigger_In pin. The scanner commences recording point cloud data upon receiving the second signal, as illustrated in Figure 3.

Time synchronization is performed through the following steps:(1)The scanner collects data based on the predetermined resolution and quality parameters. Simultaneously, the laser lens emits a synchronization pulse signal through the Trigger_Out pin at a fixed frequency corresponding to each complete rotation.(2)The inclinometer and odometer promptly collect data upon receiving the synchronization pulse signals.(3)The processor converts and integrates the data from the inclinometer and odometer into a unified dataset, appending the serial number of the synchronization pulse signal before each entry for subsequent fusion with the point cloud data.

This entire process is accomplished within 10 milliseconds. In the event of data loss or reception failure, the corresponding sensor data are set to 0 to facilitate subsequent verification and interpolation of defective data.

#### 2.1.3. Tunnel 3D Point Cloud Data Generation

The laser measurement trolley tilts inward while traversing the metro rail, attributable to its rigid connection with the scanner. This tilt induces a horizontal rotation of the section coordinate system, which is perpendicular to the direction of the trolley’s movement, leading to inaccuracies in the collected point cloud data. To mitigate this issue, a 2D inclination sensor is utilized to rectify the horizontal inclination error of the scanner, thereby enhancing data accuracy [15]. The mobile measurement system in this paper adopts the ACT926T-10 high-precision digital two-dimensional dual-axis inclination sensor. The main technical parameters of the sensor are shown in Table 1.

As illustrated in Figure 4, ideally, the point cloud data collected while the scanner is in motion should be situated within the OXZ coordinate system. However, due to the inward tilt of the scanner in conjunction with the moving platform, the coordinate system in which the actual point cloud data are collected is tilted both horizontally and laterally as a whole. Assuming that the scanner’s coordinate system is positioned within the $O_{1}X_{1}Z_{1}$ coordinate system during operation, the platform rotation angle measured by the inclinometer can be employed to correct the coordinates of all point clouds from the $O_{1}X_{1}Z_{1}$ to the OXZ coordinate system. When the platform is tilted along the track, let point C in Figure 4 represent the center of the platform, with the collected point S designated as $\left(x_{1},z_{1}\right)$ in the $O_{1}X_{1}Z_{1}$ coordinate system and (x,z) in the OXZ coordinate system. Furthermore, establishing the transformed transition coordinate system $O_{1}X_{2}Z_{2}$ allows for the calculation of (x,z) via the following coordinate transformation formula:(1)$$\alpha =\arctan\frac{x_{1}}{z_{1}}$$(2)$$x=\sqrt{x_{1}^{2}+z_{1}^{2}}*\cos\left(\frac{\pi }{2}-\gamma -\alpha \right)+D*\sin\gamma$$(3)$$z=\sqrt{x_{1}^{2}+z_{1}^{2}}*\sin\left(\frac{\pi }{2}-\gamma -\alpha \right)-D*(1-\cos\gamma )$$

Here, γ represents the tilt angle of the trolley relative to the track, as measured by a 2D inclination sensor, and D denotes the distance from the center of the platform to the center of the scanner, specifically the length of the line segment CO in Figure 4, which is a known constant.

The original point cloud data collected by the mobile laser scanning system consist of two-dimensional data composed of numerous two-dimensional section point clouds stacked together. These data only include the X and Z coordinates of the measured points within a coordinate system that is perpendicular to the scanner’s direction of motion, with no information about the scanner’s travel direction. Therefore, it is necessary to combine the original data with the time-synchronized odometer data to achieve a “stretched” representation. The original data must be “stretched” through the integration of the time-synchronized odometer data. The spiral line method is employed for the three-dimensional recovery of the original data, utilizing two adjacent mileage values as the starting and ending points of the scanning line. Assuming that the scanner operates at a constant speed during data collection, all point cloud sampling intervals are uniform. Consequently, the mileage values of the remaining points are interpolated based on the starting and ending mileage values of the scanning line, as illustrated in Figure 5.

A total of N scanning lines are established, numbered from 1 to N, with each scanning line containing n scanning points. The coordinates of the jth point $p_{ij}$ on the ith scanning line are defined as $\left(x_{ij},y_{ij},z_{ij}\right)$. The odometer data and scanning line data correspond one to one based on the sampling moments, resulting in N mileage data points represented as $M_{1},M_{2},\ldots M_{i}\ldots ,M_{N}$. Subsequently, the Y coordinate of any point $p_{ij}$ can be calculated using the following formula:(4)$$y_{ij}=\left(M_{i+1}-M_{i}\right)*\frac{j}{n}$$

### 2.2. Research on Point Cloud Data Pre-Processing Algorithms for Metro Shield Tunnels

The integrated mobile laser measurement system generates high-density point cloud data of the tunnel surface, but raw data cannot directly calculate tunnel deformation. Pre-processing is required, including removing discrete points, determining the median axis, and separating the track from the roadbed. Based on this, mileage positioning is achieved through ring seam identification, providing relative positional information for the tunnel point cloud.

#### 2.2.1. Tunnel 3D Point Cloud Data Generation

The original point cloud data contain a significant number of noise points, including outlier noise points that are distant from the main point cloud and disturbance points that are in close proximity to the tunnel wall. The presence of noise points can interfere with the subsequent processing of the tunnel point cloud; thus, a Statistical Outlier Removal (SOR) filtering algorithm is applied to remove the outlier noise points. Assuming that the distances of all points follow a Gaussian distribution, the input point cloud is traversed to calculate the distance $d_{i}$ of each point to its neighborhood k. Let the coordinates of the ith point $P_{i}$ in the point cloud be $\left(x_{i},y_{i},z_{i}\right)$; the distance from this point to any point $P_{m}(x_{m},y_{m},z_{m})$ is calculated as follows:(5)$$d_{i}=\sqrt{{\left(x_{i}-x_{m}\right)}^{2}+{\left(y_{i}-y_{m}\right)}^{2}+{\left(z_{i}-z_{m}\right)}^{2}}$$

Calculate the mean μ, standard deviation σ, and set the threshold $d_{max}$:(6)$$\mu =\frac{1}{n}\sum_{i=1}^{n}d_{i}y$$(7)$$\sigma =\sqrt{\frac{1}{n}\sum_{i=1}^{n}{\left(d_{i}-\mu \right)}^{2}}$$

The threshold value is typically defined as an integer multiple of the standard deviation. If $d_{i}<d_{max}$, the point is classified as an inner point and retained; conversely, if $d_{i}\geq d_{max}$, it is classified as an outlier noise point and removed.

Upon completion of outlier removal, the central axis of the tunnel is determined. This central axis represents the attitude and direction of the tunnel. Given the substantial amount of data in the tunnel point cloud, the extraction of the central axis is transformed from three-dimensional space to two-dimensional space to enhance algorithm efficiency. The tunnel point cloud is projected onto the YOZ and XOY planes, allowing for the derivation of centerlines on these two 2D planes. These two 2D centerlines are then combined to represent the three-dimensional axis of the tunnel. The specific steps are as follows:(1)Project the tunnel point cloud onto the YOZ and XOY planes, defining the ith point $P_{i}$ in the point cloud with coordinates $\left(x_{i},y_{i},z_{i}\right)$. The coordinates after projection are denoted as $\left(x_{P},y_{P},z_{P}\right)$, and the general equation of the projection plane is expressed as(8)$$Ax+By+Cz=0y_{P}=\frac{B}{A}\left(x_{P}-x_{i}\right)+y_{i},z_{P}=\frac{C}{A}\left(x_{P}-x_{i}\right)+z_{i}$$

The line connecting the projected point and the point to be projected must be perpendicular to the plane of projection, ensuring that $y_{P}$ and $z_{P}$ satisfy the following conditions:(9)$$y_{P}=\frac{B}{A}\left(x_{P}-x_{i}\right)+y_{i},z_{P}=\frac{C}{A}\left(x_{P}-x_{i}\right)+z_{i}$$

Conditions for solvability are(10)$$\left\{\begin{array}{l} x_{P}=\frac{\left(B^{2}+C^{2}\right)x_{i}-A\left(By_{i}+Cz_{i}+D\right)}{A^{2}+B^{2}+C^{2}}\\ y_{P}=\frac{\left(A^{2}+C^{2}\right)y_{i}-B\left(Ax_{i}+Cz_{i}+D\right)}{A^{2}+B^{2}+C^{2}}\\ z_{P}=\frac{\left(A^{2}+B^{2}\right)z_{i}-C\left(Ax_{i}+Cy_{i}+D\right)}{A^{2}+B^{2}+C^{2}} \end{array}\right.$$

(2)Define the boundaries of the projected point cloud on the YOZ and XOY planes and resample the 2D midline points. For the YOZ plane, the point cloud Q is segmented into n subsets $Q_{1},Q_{2},\ldots ,Q_{n}$ using a specified step size. The coordinate extremes $y_{min}^{i},y_{max}^{i},z_{min}^{i},z_{max}^{i}$ in each segment are averaged to form discrete median point sets $\left\{Q_{1},Q_{2},\ldots ,Q_{n}\right\}$. The tunnel spans 240–300° counterclockwise from the X-axis in the Cartesian system, leaving this range without tunnel wall points. Thus, the centerline coordinates are determined using the following equation:(11)$$Q_{iy}=\frac{y_{\min\;}^{i}+y_{\max\;}^{i}}{2},Q_{iz}=\frac{z_{\min\;}^{i}+z_{\max\;}^{i}}{2}-\frac{2-\sqrt{3}}{4}*R$$(3)Employ weighted least squares to fit a straight line to the resampled two-dimensional midline points in both the YOZ and XOY planes. For instance, in the YOZ plane, let the equation of the line $l_{1}$ be expressed as $z=a_{1}y+b_{1}$. This problem can be reformulated by finding the parameters $a_{1}$ and $b_{1}$ such that $D_{sum}=\sum_{i=1}^{n}\left[z_{i}-(a_{1}y_{i}+b_{1})\right]$ is minimized. The calculation can be performed as follows:(12)$$a_{1}=\frac{n\sum y_{i}z_{i}-\sum y_{i}\sum z_{i}}{n\sum y_{i}^{2}-{\left(\sum y_{i}\right)}^{2}},b_{1}=\frac{n\sum z_{i}-a_{1}\sum y_{i}}{n}$$(4)By employing the same method, the equation of the two-dimensional midline $l_{2}$ in the XOY plane can be derived as $x=a_{2}y+b_{2}$. Subsequently, the spatial midline L can be jointly represented by $l_{1}$ and $l_{2}$:(13)$$L:\frac{z-b_{1}}{a_{1}}=y=\frac{x-b_{2}}{a_{2}}$$

Non-tunnel wall points negatively impact section fitting, so after removing noise points, separating the tunnel from the roadbed is necessary. The isolated tunnel wall is then used for deformation calculations. This study identifies the tunnel bed using its angular distribution (240–300°) relative to the X-axis in a shield tunnel coordinate system [16]. All points are analyzed to establish a 2D coordinate system for the section. For the ith point $P_{i}(x_{i},y_{i},z_{i})$, the origin of the section’s coordinate system is defined as $O_{c}(x_{O_{c}},z_{O_{c}})$. The angle $A_{b}$ between the line connecting each point to the origin and the X-axis is calculated as follows:(14)$$A_{b}=\left\{\begin{array}{l} \arctan\left(\frac{z_{i}-z_{O_{c}}}{x_{i}-x_{O_{c}}}\right)\;if\;x_{i}>x_{O_{c}},z_{i}>z_{O_{c}}\\ \arctan\left(\frac{z_{i}-z_{O_{c}}}{x_{i}-x_{O_{c}}}\right)+2\pi \;if\;x_{i}>x_{O_{c}},z_{i}>z_{O_{c}}\\ \arctan\left(\frac{z_{i}-z_{O_{c}}}{x_{i}-x_{O_{c}}}\right)+\pi \;if\;x_{i}<x_{O_{c}} \end{array}\right.$$

When 240°<A<300°, the point is classified as a roadbed point and subsequently removed.

#### 2.2.2. Mileage Localization of Tunnel Point Clouds Based on Bolt Hole Extraction

The point cloud data collected by the mobile laser system lack accurate absolute position information, so it is very important to calibrate the mileage of the collected point cloud. The ring width of the shield subway tunnel is fixed, and there are ring gaps between the rings. There are bolt holes distributed around the ring gaps in a certain pattern, as shown in Figure 6. Mileage calibration can be achieved by extracting the bolt holes to locate the ring gaps.

During the operational period, the pressure on the sides of the metro tunnel is usually less than the pressure at the top. Therefore, this paper focuses on extracting the bolt holes at the tunnel’s top for mileage calibration. A 2D Cartesian coordinate system is established for the section, with the coordinates of the ith point $P_{i}(x_{i},y_{i},z_{i})$ and the origin $O_{ci}(x_{O_{ci}},z_{O_{ci}})$. The distance between $P_{i}$ and $O_{ci}$ is $l_{i}$, and the angle $A_{i}$ relative to the X-axis is also calculated. If $l_{i}>R$ and $A_{i}\in \left[\frac{5}{36}\pi ,\frac{2}{9}\pi \right]\cup \left[\frac{5}{18}\pi ,\frac{13}{36}\pi \right]$, the bolt holes at the top are identified [17]. This process continues until all points are evaluated. After extracting the bolt holes, the K-Means++ algorithm is used to cluster the bolt holes based on their mileage direction [18]. K-Means++ optimizes initial centroid selection, speeding up convergence compared to traditional K-Means.

Based on the distribution characteristics of the tunnel bolt holes illustrated in Figure 6, each ring contains two columns of bolt holes. The point cloud clusters corresponding to these two columns in the same ring are denoted as $P_{i}^{1}$ and $P_{i}^{2}$, respectively. These clusters can be grouped along the direction of the bolt holes, specifically utilizing the Y coordinate information to identify both the ring seams and the center position of each ring. Define the Y coordinate of the ring seam as $H_{y}$, and the Y coordinate at the center of each ring as $C_{y}$. Let the number of points in $P_{i}^{1}$ and $P_{i}^{2}$ be $n_{i}^{1}$ and $n_{i}^{2}$, respectively, with the average Y coordinates represented as $y_{i}^{1}$ and $y_{i}^{2}$. The values of $H_{y}$ and $C_{y}$ can then be calculated using the following formulas:(15)$$H_{y}=\frac{y_{i}^{2}+y_{i+1}^{1}}{2}\;C_{y}=\frac{y_{i}^{1}+y_{i}^{2}}{2}$$

### 2.3. Deformation and Encroachment Detection of Tunnel Structure Based on Moving Laser Point Cloud

#### 2.3.1. Tunnel Section Extraction and Disturbance Point Removal

Tunnel laser point cloud data are characterized by their massive volume, which can hinder the efficiency of deformation detection in actual projects. To address this issue, the tunnel cross-section point cloud is utilized as a foundational dataset for deformation detection, ensuring that the algorithms employed can operate with high efficiency.

Section extraction relies on the central axis line, as illustrated in Figure 7. Using the established central axis of the tunnel space, a movable normal plane $l_{i}$ is constructed, defining the central axis line L:(16)$$L:\frac{z-b_{1}}{a_{1}}=y=\frac{x-b_{2}}{a_{2}}$$

A perpendicular moving normal plane is constructed along the tunnel axis, which intercepts the tunnel section point cloud at various mileage points as it moves in the tunnel direction. The mileage value $y_{i}$ at the normal plane is designated as $l_{i}$, and the analytical formula for $l_{i}$ can be calculated based on the axis:(17)$$l_{i}:a_{2}\left(x-b_{2}\right)-a_{1}\left(y-b_{1}\right)=0$$

Different sections are selected for calculation based on specific deformation detection needs. To detect tunnel section convergence, as illustrated in Figure 7, the section at the center of each ring is chosen for convergence analysis. A buffer zone of Δd is established on both sides of the solving area, with the normal plane at the center designated as $l_{m}$ and the mileage value as $y_{m}$. Point cloud data within the range $\left[y_{m}-\Delta d,y_{m}+\Delta d\right]$ are projected onto the normal plane $l_{m}$. To detect inter-ring misalignment, the ring joint location is chosen as the reference section. The normal plane at the ring joint is designated as $l_{n}$, with point clouds in the intervals $\left[y_{n}-\Delta d,y_{n}\right]$ and $\left[y_{n},y_{n}+\Delta d\right]$ projected onto the $l_{n}$ plane. This approach allows for obtaining point clouds of two sections with the same mileage values for misalignment detection. For tunnel limit detection, as encroachment is not related to tunnel deformation, the previous section extraction method is unsuitable. Instead, a point-by-point construction of the section coordinate system is employed for calculations.

The section extracted using the aforementioned method still contains numerous perturbation points near the tunnel wall, including pipelines, cables, contact nets, and bolt holes. These points can adversely affect the accuracy of subsequent section fitting, leading to errors in solving the deformation parameters of the tunnel section. Given that the deformation section of a shield tunnel resembles an ellipse, this paper employs the RANSAC ellipse fitting algorithm, which demonstrates superior noise resistance, for the removal of disturbance points. The algorithm proceeds as follows:(1)Initialize the maximum number of iterations, the initial set of fitted points, and the distance threshold between the points and the model while randomly selecting the initial optimal ellipse.(2)Select the inner points and perform direct least squares fitting based on these points to obtain the ellipse parameters.(3)Calculate the sum of the densities of all points on the ellipse to determine the optimal ellipse, along with the maximum number of iterations for the fitting process.(4)Repeat steps (2) and (3) until the fitting count exceeds the maximum number of iterations. Upon completion, obtain the final ellipse parameters and remove the section points that do not meet the set distance threshold.

#### 2.3.2. Convergence Deformation Detection of Shield Tunnel

During metro tunnel operations, the deformation of each point on the tunnel tube sheet relative to the central position is referred to as tunnel section convergence, influenced by factors such as ground building loads, large-scale surrounding construction, and vibrations from train operations. The shape of the section post-deformation is typically elliptical. To determine the tunnel convergence value from point cloud data, this paper fits the point cloud data to an ellipse, addressing ellipticity, tube sheet convergence, and full section convergence.

This paper employs a direct minimum absolute deviation ellipse fitting method that minimizes algebraic distance [19], defined by the following quadratic curve equation:(18)$$F(M,N)=Ax^{2}+Bxz+Cz^{2}+Dx+Ez+F=0$$
where any point $p_{i}(x_{i},z_{i})$ on the section, the number of points is n, $M_{i}=\left[\begin{array}{cccccc} x_{i}^{2} & x_{i}z_{i} & z_{i}^{2} & x_{i} & z_{i} & 1 \end{array}\right]$, $N={\left[\begin{array}{cccccc} A & B & C & D & E & F \end{array}\right]}^{T}$, $\left|M_{i}N\right|$ is the algebraic distance between $p_{i}$ and the model curve F(M,N)=0, and the sum of the algebraic distances of all points is(19)$$F(M,N)=\sum_{i=1}^{n}\left|M_{i}N\right|$$

Since the cross-section point-fitting model is an ellipse, it should satisfy $B^{2}-4AC<0$; otherwise, the fitting result may be hyperbolic or parabolic. To facilitate the solution, a subset $4AC-B^{2}=1$ is taken, in which the matrix form is $N^{T}CN=1$, and(20)$$C=\left[\begin{array}{llllll} 0 & 0 & 2 & 0 & 0 & 0\\ 0 & -1 & 0 & 0 & 0 & 0\\ 2 & 0 & 0 & 0 & 0 & 0\\ 0 & 0 & 0 & 0 & 0 & 0\\ 0 & 0 & 0 & 0 & 0 & 0\\ 0 & 0 & 0 & 0 & 0 & 0 \end{array}\right]$$

This means that minimizing the target elliptic model leads to the following matrix formulation for the problem:(21)$$\left\{\begin{array}{c} \underset{N}{\min}{\left\|MN\right\|}_{1}\\ s.t.\;N^{T}CN=1 \end{array}\right.$$

Introduction of Lagrange operators λ:(22)$$\underset{N}{\arg\;\min}\{{\left\|MN\right\|}_{1}+\lambda (N^{T}CN-1)\}.$$

The objective of ellipse fitting is to identify the optimal N that minimizes the aforementioned equation. This is achieved using the Bregman method, which introduces an auxiliary variable s=MN to transform the unconstrained minimization problem into a constrained one:(23)$$\left\{\begin{array}{l} \underset{N}{\arg\;\min}\{{\left\|s\right\|}_{1}+\lambda (N^{T}CN-1)\}.\\ s.t.\;s=MN. \end{array}\right.$$

Adding a penalty function term for the solution reduces the above equation to(24)$$\underset{N,s}{\arg\;\min}\{{\left\|s\right\|}_{1}+\frac{\mu }{2}{\left\|s-MN\right\|}_{2}^{2}+\lambda (N^{T}CN-1)\}$$
where μ is a positive penalty parameter. Finally, the problem is converted into two subproblems by applying the split Bregman iteration with the Bregman variable t strictly enforcing the constraints:(25)$$\underset{N,s,t}{\arg\;\min}\{{\left\|s\right\|}_{1}+\frac{\mu }{2}{\left\|s-MN-t\right\|}_{2}^{2}+\lambda (N^{T}CN-1)\}$$(26)$$t^{k+1}=t^{k}+MN^{k+1}-s^{k+1}$$

The above formula can be further simplified into two sub-problems, among which the N-related problems are:(27)$$\underset{N}{\arg\;\min}\{\frac{\mu }{2}{\left\|s-MN-t\right\|}_{2}^{2}+\lambda (N^{T}CN-1)\}$$

The above equation is a least squares problem that leads to a closed solution:(28)$$N^{k+1}=\frac{\mu M^{T}(s^{k}-t^{k})}{\mu M^{T}M+2\lambda C}$$

s-related issues for(29)$$\underset{s}{\arg\;\min}\{{\left\|s\right\|}_{1}+\frac{\mu }{2}{\left\|s-MN-t\right\|}_{2}^{2}\}$$

The above equation can be solved using the standard soft threshold formula:(30)$$s^{k+1}=shrink(MN^{k+1}+t^{k},\frac{1}{\mu }),\;shrink(x,r)=\frac{x}{\left|x\right|}\max(\left|x\right|-r,0)$$

Based on the ellipse parameters sought, the ellipse center coordinates $\left(X_{c},Z_{c}\right)$, the lengths of the long and short axes $R_{x}$, $R_{z}$ and the deflection angle θ can be calculated:(31)$$\left\{\begin{array}{l} X_{c}=\frac{\left(BE-2CD\right)}{4AC-B^{2}}\\ Z_{c}=\frac{\left(BD-2AE\right)}{4AC-B^{2}}\\ R_{x}=\sqrt{\frac{2(AX_{c}^{2}+CZ_{c}^{2}+BX_{c}Z_{c}-1)}{A+C-\sqrt{{\left(A-C\right)}^{2}+B^{2}}}}\\ R_{z}=\sqrt{\frac{2(AX_{c}^{2}+CZ_{c}^{2}+BX_{c}Z_{c}-1)}{A+C+\sqrt{{\left(A-C\right)}^{2}+B^{2}}}}\\ \theta =\frac{1}{2}\tan^{-1}\frac{B}{A-C} \end{array}\right.$$

Common indices for measuring the deformation of circular shield tunnels include horizontal diameter and ellipticity. Following deformation, the direction of the long axis of the tunnel is typically not aligned with the horizontal orientation of the section. In this paper, for convergence calculations, the length of the long half-axis obtained from the ellipse fitting of the resolved section is chosen as the deformation value, along with the difference from the design radius. Ellipticity is calculated as the ratio of the difference between the maximum and minimum diameters to the design radius, expressed in thousandths:(32)$$E=\frac{R_{x}-R_{z}}{R}*1000‰$$
where E is the ellipticity, $R_{x}$ and $R_{z}$ are the lengths of the long and short axes, and R is the design radius.

#### 2.3.3. Shield Tunnel Inter-Ring Misalignment Detection

When uneven external forces act on the tunnel, relative displacement occurs at the tube sheet joints, causing ring misalignment. Transverse misalignment refers to the displacement of internal splicing blocks within a ring, while radial (inter-ring) misalignment is the misalignment between adjacent rings perpendicular to the tunnel axis. Inter-ring misalignment detection uses point clouds from both sides of the ring seam, which are fitted to ellipses with the origin at the fitting center. The point clouds are divided into 1° segments across the 0–360° range, and the average geometric distance from each point to the corresponding ellipse center is calculated. The results for corresponding regions in both sections are compared to determine inter-ring misalignment. After cross-section filtering, point cloud density may become non-uniform, with some angular ranges lacking data due to noise. When calculating misalignment, if any angular section lacks data, it is skipped, and the misalignment for that section is recorded as zero. Let the distances from the two detection points to the centers of their ellipses be denoted by sets L and R, with inter-ring misalignment quantities represented by M:(33)$$M=\{L_{1}-R_{1},L_{2}-R_{2},\cdots L_{i}-R_{i},\cdots ,L_{360}-R_{360}\}$$

#### 2.3.4. Tunnel Locomotive Limit Encroachment Detection

The Metro Tunnel Vehicle Limit refers to the limiting cross-sectional profile that is perpendicular to the centerline of the track and serves to restrict the external dimensions of the rolling stock. The calculation of the vehicle—whether empty or loaded—moving in a flat, straight line on the track is based on the specified speed while accounting for vehicle and track tolerance values, wear and tear, elastic deformation, vehicle vibrations, and potential failures in one or both suspension systems. These factors contribute to lateral and vertical dynamic offsets of vehicle components, resulting in the formation of a dynamic envelope. This envelope must be expressed within the base coordinate system, ensuring that the boundaries do not encroach upon any installation equipment.

The Metro model equipment limit can be conceptualized as a planar closed area. Any such area can be represented as an irregular polygon, allowing the limit detection problem to be transformed into a judgment of the positional relationship between points and polygons. The relationship between points and a closed polygon can be categorized into three cases: points located inside the polygon, points outside the polygon, and points located at the vertices or on the edges of the polygon, as illustrated in Figure 8. Common methods for determining the positional relationship between points and polygons include the angle-based method, area-based method, topological mapping method, and the ray method. Compared to other methods, the ray method features simpler decision rules and lower computational complexity. Subsequent experiments have demonstrated that this method effectively identifies intrusions. Therefore, this paper employs the ray method for detecting locomotive limit intrusions [20].

Let the vertices of the polygon be denoted as $E=\{E_{1},E_{2},...E_{i}...,E_{n}\}$. The set E represents an ordered collection of points that are sequentially connected to form a restricted polygon. Consider the point to be measured as $m(x_{m},z_{m})$. A horizontal ray mn is drawn to the right of point m. Let the number of intersections between the ray and the polygon be denoted as $I_{n}$, and define the positional relationship between the point and the polygon as $I_{pos}$. The geometric properties of the vector cross product are employed to determine the positional relationship between the measured point and the closed polygon, defined as follows:(34)$$V_{1}=mn\times mE_{i},V_{2}=mn\times mE_{i+1},V_{3}=E_{i}E_{i+1}\times E_{i}m,V_{4}=E_{i}E_{i+1}\times E_{i}n$$

Traverse each edge of the sealed polygon sequentially and set the ith edge coordinate poles to be $X_{min},\;{\;Z}_{min},{\;X}_{max},\;and{\;Z}_{max}$; the specific judgment method is
(1)If $V_{1}V_{2}>0$ or $V_{3}V_{4}>0$, $I_{pos}=0$.(2)If $V_{1}V_{2}<0$ and $V_{3}V_{4}<0$, $I_{n}=I_{n}+1$.(3)If one of $V_{1}$ and $V_{2}$ is 0, the ray mn must cross one of the vertices of the polygon and determine whether the two points adjacent to that vertex are on either side of $E_{i}E_{i+1}$. If, on both sides, there is $I_{n}=I_{n}+1$. If on the same side, $I_{n}$ is unchanged.(4)If $V_{1}V_{2}<0$ and $V_{3}=0$, $I_{pos}=0$.(5)If $V_{1}V_{2}=0$ and $V_{3}V_{4}=0$, the ray mn must be colinear with $E_{i}E_{i+1}$. If $X_{min}<x_{m}<X_{max}$, the point to be measured lies on $E_{i}E_{i+1}$ then $I_{pos}=0$. If $x_{m}<X_{min}$, determine whether the two adjacent vertices $E_{i}$ and $E_{i+1}$ of $E_{i}E_{i+1}$ are on both sides of mn, if on both sides then there is $I_{n}=I_{n}+1$; if on the same side then we have $I_{n}=0$.

If the final calculation result $I_{n}$ is odd then $I_{pos}=1$; if it is even then $I_{pos}=-1$, and the encroachment limit point judgment is determined according to the following formula:(35)$$I_{pos}\left\{\begin{array}{c} =1,the\;point\;to\;be\;measured\;is\;inside\;a\;polygon\\ =0,the\;point\;to\;be\;measured\;is\;at\;a\;polygon\;vertex\;or\;on\;an\;edge\\ =-1,the\;point\;to\;be\;measured\;is\;outside\;a\;polygon \end{array}\right.$$

When the point to be detected is inside a polygon, at a polygon vertex, or on an edge, the point to be detected is determined to be an encroachment point.

## 3. Results

The measured point cloud data from the Nanjing Metro tunnel are utilized to evaluate the effectiveness of the algorithms proposed in this paper, including tunnel point cloud pre-processing, section convergence analysis, inter-ring misalignment detection, and vehicle limit intrusion assessment. The experimental data consist of the measured point cloud data from a segment of the Nanjing Metro Line 2 uplink constructed using the shield method. The design diameter of the tunnel’s tube sheet is 5.50 m, with a total length of approximately one kilometer. The data acquisition process encompasses several stages, including parameter configuration, mobile scanning, external data verification, and data backup. Given the extensive volume of point cloud data, 37 ring slices were selected for analysis to effectively demonstrate the processing results of the algorithm discussed in this chapter.

### 3.1. Tunnel Point Cloud Pre-Processing and Mileage Calibration

Statistical filtering is influenced by both the number of queried near-neighbor points in the distance statistics and the threshold value. In this experiment, the number of neighboring points queried was set to 70, and the threshold was defined as twice the standard deviation. These values were determined as optimal through extensive experimental testing and comparison. Furthermore, a comparison with manually filtered results revealed a high level of consistency, confirming that the selected parameters effectively removed noise while preserving valid points. The denoising effect is illustrated in Figure 9, demonstrating a significant improvement, with approximately 1500 noise points successfully removed.

Following the removal of outlier points from the tunnel, 2 two-dimensional midlines are employed to represent the three-dimensional midline of the tunnel. The results obtained from solving and extracting the midline in the experimental area are illustrated in Figure 10:

Where the three-dimensional midline is computed as follows:$$L:\frac{z-1.19604}{0.00114409}=y=\frac{x-0.077105}{-0.002262}$$

After identifying the central axis, the tunnel must be separated from the roadbed using the empirical shield tunnel angle range of 150° to 210°, where 0° corresponds to the vertical upward direction. The resulting removal effects are shown in Figure 11.

Upon completion of the point cloud data pre-processing, the bolt holes located at the top of the tunnel were extracted and clustered using the KMeans++ algorithm; the results of extraction and clustering are presented in Figure 12:

Utilizing the clustering results, the coordinate ranges of similar bolt holes were calculated to ascertain the mileage values for the locations of the ring center and the ring seam within the tunnel. The marking results are illustrated in the grayscale diagram shown in Figure 13:

### 3.2. Tunnel Convergence Detection and Analysis

Utilizing the convergent deformation detection section point cloud as the calculation unit, the solved tunnel central axis is employed to construct a moving normal plane. Combined with the results from the tunnel ring seam and the ring center mileage calculations, each ring is divided into five uniformly spaced sections along the tunnel direction, which serve as cross-sections for convergence solving. The RANSAC algorithm is employed for ellipse fitting with a set distance threshold; points that do not meet this threshold are identified as noise and subsequently removed, while tunnel wall points are retained for further ellipse fitting. The effects of section extraction and denoising are illustrated in Figure 14, where red points indicate retained tunnel wall points and white points represent removed noise points.

Following the denoising of the section point cloud, the convergent solution is performed. For each ring, five sections are utilized for convergence solving, all of which undergo elliptical fitting. The section exhibiting the smallest standard deviation of the long axis is selected as the optimal section for convergence solving of that ring slice. The results of some ring slice solutions are presented in Table 2:

By correlating the lengths of the short and long axes of the fitted ellipse with the design radius of the tunnel, the ellipticity and convergence values of the tunnel section can be computed. The design radius of the circular shield tunnel in the experimental area is 2.75 m, and the results for the ellipticity and convergence values of selected tunnel tubes are presented in Table 3.

Following the ellipse fitting of the tunnel section, to obtain the deformation for each angle, the entire section is converged and solved by dividing the range into 360 intervals from 0° to 360° based on the clockwise orientation of the section coordinate system. Each point within these intervals is traversed, calculating the difference between the distance of each point from the center of the section and the design radius. The average difference across the intervals is then considered as the deformation corresponding to each angle. The absence of points in certain areas allows for the calculation of the deformation values at corresponding angles using the fitted elliptic analytical equation. The results of the full-section solution are illustrated in Figure 15.

To verify the accuracy of the solving results, the convergence value calculations presented in this paper are compared with those obtained from the NJCK-Track One system, as illustrated in Figure 16. The results indicate that the discrepancies between this paper’s findings and those of the NJCK-Track One system are minimal. Specifically, the differences for 37 ring slices are within 3 mm, with an average difference of 1.3 mm across all ring slices, demonstrating strong consistency between the converged results of this paper and those of the NJCK-Track One system.

### 3.3. Tunnel Convergence Detection and Analysis

For inter-ring misalignment detection, the cross-section at the ring seam is selected as a benchmark, with two comparative sections chosen approximately 10 cm before and after the seam. Similarly to the convergence detection, the extracted sections undergo RANSAC filtering to remove perturbation points. The denoised sections are then ellipse-fitted to obtain the centroid coordinates, long and short axes, and angle of deflection. Some of the filtered comparative sections used for inter-ring misalignment detection and solving are illustrated in Figure 17.

All points in the two sections are traversed, and the section is divided into 360 intervals ranging from 0° to 360°. The average distance from each point to the section center is calculated for each interval, and the differences in distances between the two sections in the same angular intervals are used to determine the inter-ring misalignment in the corresponding angular regions. In noisy areas, such as those with pipelines and cables, if a filtered area does not exist in a region, the misalignment is not calculated for that area. Some of the inter-ring misalignment results are shown in Table 4.

Full-section deformation statistics are conducted to assess the misalignment between individual rings, specifically calculating the misalignment between adjacent pipe sheets at each angle. The results of the inter-ring misalignment calculations are presented in Figure 18, where the maximum misalignment between rings 334 and 335 occurs at 350°.

### 3.4. Vehicle Limit Intrusion Detection and Analysis

In the vehicle limit intrusion detection process, the original CAD file of the locomotive is converted into a 3D point cloud file by identifying point and line elements. Subsequently, the point cloud representing the locomotive profile is positioned at the section to be tested based on the mileage where the section is detected and the corresponding section center coordinates. The red points in Figure 19 represent the characteristic points of the locomotive contour. Given that the experimental area is a tunnel currently in service, the scanned point cloud is assessed to be free of any infringing points. Therefore, random point cloud data added to the original data file are utilized to validate the infringement detection algorithm. As illustrated in Figure 19, the numbered positions 1 to 4 represent four distinct relationships between the point to be measured and the closed polygon: the point lies at the polygon vertex, the point resides inside the polygon, the point is located outside the polygon, and the point is situated on the polygon edge.

The section points are systematically traversed, and encroachment points are identified utilizing the limit method. The identification results are presented in Figure 20, where the red points indicate the encroachment locations.

Following the detection of encroachment at the section points, the locations of the encroachment points, along with their respective encroachment lengths, are calculated and presented in Table 5.

## 4. Discussion

The mobile laser measurement system developed in this study demonstrated strong performance during the Xuzhou Metro field experiment, confirming its practicality for tunnel structure inspection. The system successfully integrated multiple sensors with precise time synchronization, while the point cloud pre-processing and deformation detection algorithms showed high accuracy and efficiency in detecting tunnel convergence, inter-ring misalignment, and limit intrusion. Compared to traditional inspection methods, this system offers significant advantages, including a high level of automation, rapid data acquisition and processing, and enhanced detection accuracy. These features not only improve inspection efficiency but also reduce manual intervention and labor costs.

However, the tunnel processing algorithm developed here primarily targets circular shield tunnels without fully considering the needs of other tunnel shapes. Additionally, while the system achieves high relative accuracy between tunnel segments, it does not fully capture the true three-dimensional shape of the tunnel. Future improvements may involve integrating sensors such as IMUs and structured light sensors. Moreover, to identify defects like block falls and cracks, additional sensors such as linear array cameras will be required to enhance the system’s functionality.

Overall, this study presents an effective technical solution for automated deformation detection in subway tunnels. With further optimization of the algorithm, improved environmental adaptability, and the integration of additional sensors, the system holds great potential for broader engineering applications.

## 5. Conclusions

This paper presents the development of a mobile laser measurement system designed for the deformation detection of metro tunnel structures alongside a pre-processing algorithm for tunnel point cloud data and a deformation detection methodology. The system integrates various modules, including the track inspection car, scanner, odometer, and inclinometer, with a design based on a multi-sensor time synchronization scheme utilizing TTL mode. By employing data from the inclinometer and odometer, the system achieves point cloud data coordinate correction and three-dimensional recovery. Field tests conducted in Xuzhou Metro demonstrate that the system meets the requirements for routine underground structural health inspections. To facilitate deformation detection of tunnel structures using point cloud data, a pre-processing algorithm has been developed, which encompasses discrete point removal, central axis determination, roadbed removal, and mileage positioning of the tunnel point cloud. For detecting tunnel convergence and inter-ring misalignment, distinct sections are extracted based on refined filtering and fitting of the section point cloud, enabling accurate solution completion. The ray method is employed for encroachment detection, facilitating the identification of encroachment points and the calculation of encroachment information. The mobile laser measurement system developed in this study offers a methodological reference for automated deformation detection in metro tunnels, significantly enhancing inspection efficiency.

## Figures and Tables

**Figure 1 sensors-25-00356-f001:**
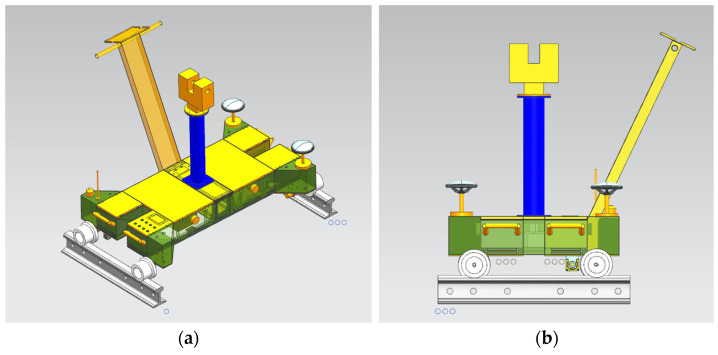
Model diagram of mobile laser measurement system. (**a**) Overall diagram of the measurement system model; (**b**) measurement system model side view.

**Figure 2 sensors-25-00356-f002:**
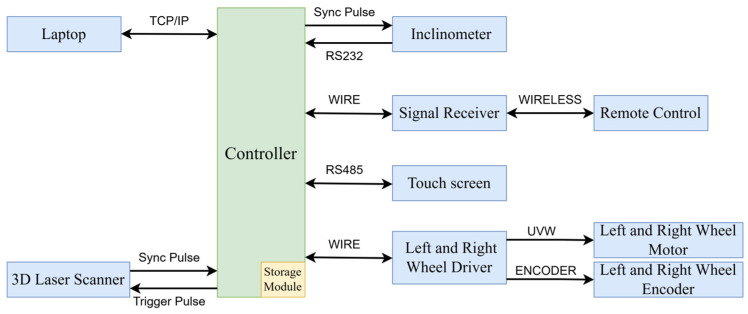
Overall communication plan of the system.

**Figure 3 sensors-25-00356-f003:**
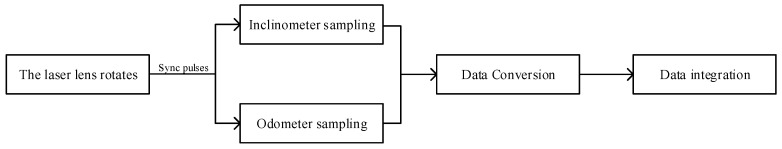
Time synchronization process.

**Figure 4 sensors-25-00356-f004:**
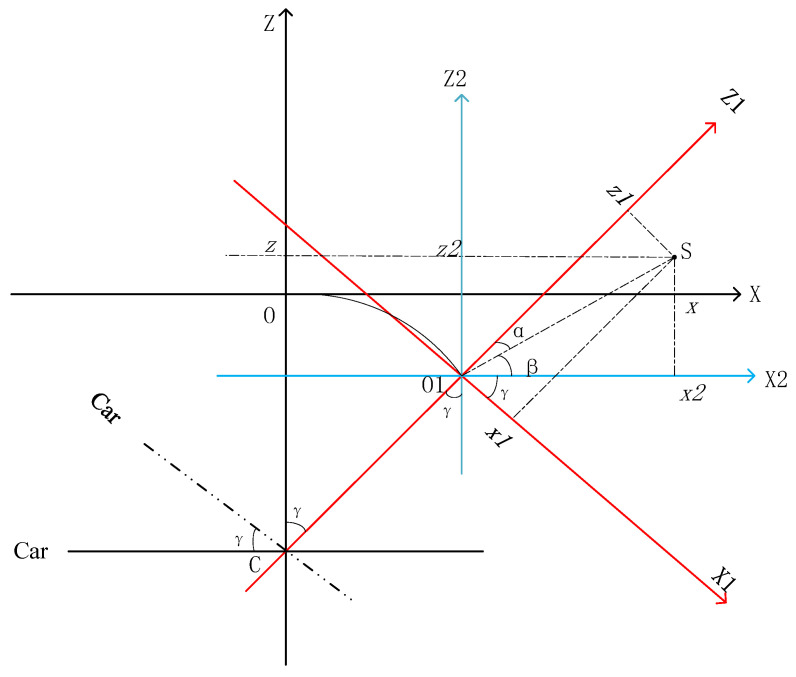
Schematic diagram of the inclined section coordinate system.

**Figure 5 sensors-25-00356-f005:**
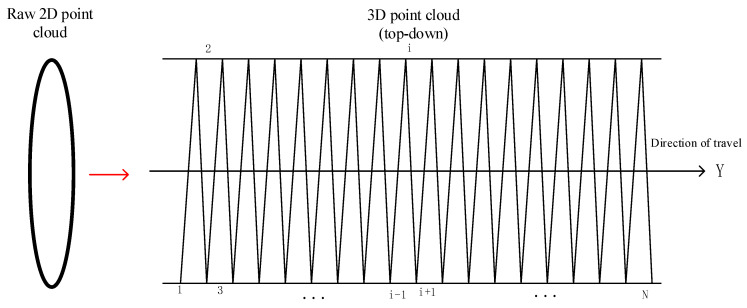
Spiral line method for the 3D reconstruction of the tunnel point cloud.

**Figure 6 sensors-25-00356-f006:**
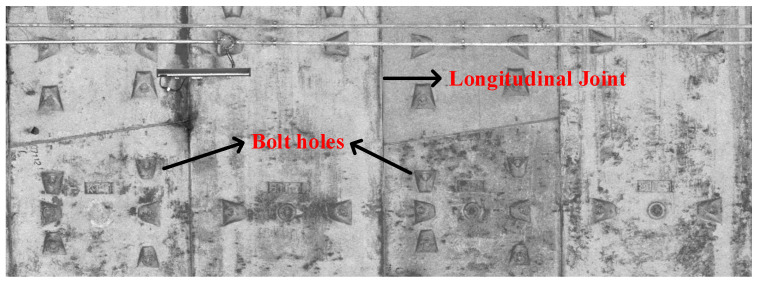
Distribution of bolt holes and longitudinal joint characteristics.

**Figure 7 sensors-25-00356-f007:**
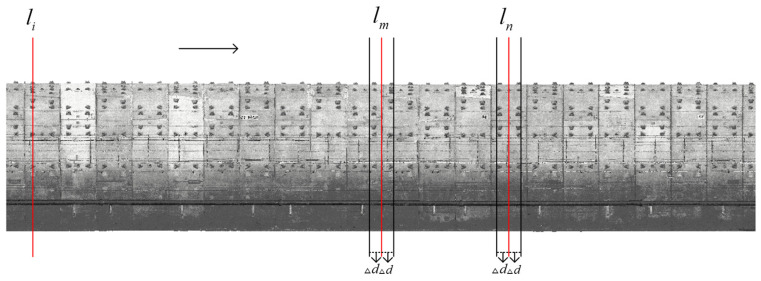
Schematic diagram of tunnel cross-section extraction.

**Figure 8 sensors-25-00356-f008:**
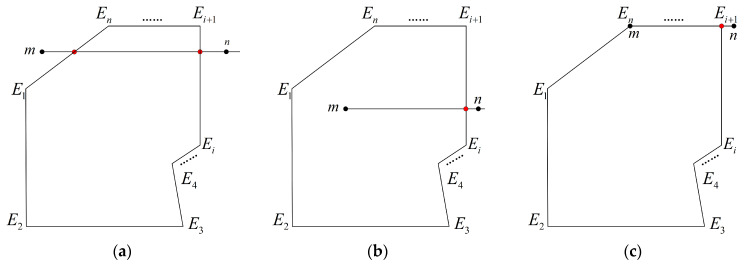
Relationship between points and polygon position: (**a**) points outside the polygon; (**b**) the point inside the polygon; (**c**) points on polygon vertices or edges.

**Figure 9 sensors-25-00356-f009:**
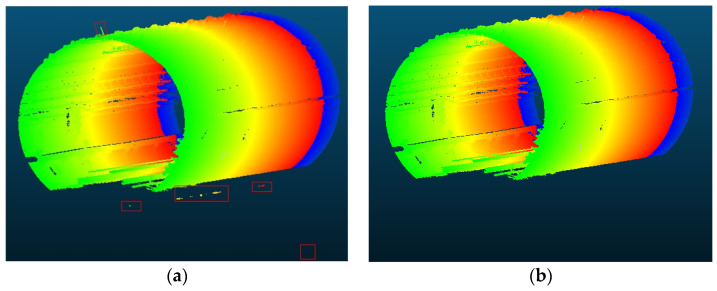
Removal effect of outlier noise points: (**a**) before removing outlier noise points; (**b**) after removing outlier noise points.

**Figure 10 sensors-25-00356-f010:**
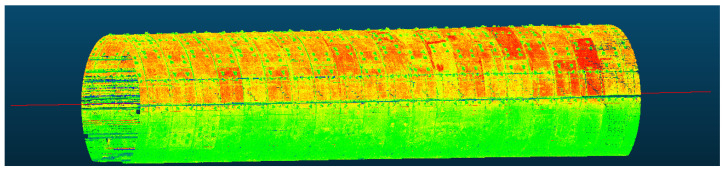
Resampling results of the tunnel 3D axis.

**Figure 11 sensors-25-00356-f011:**
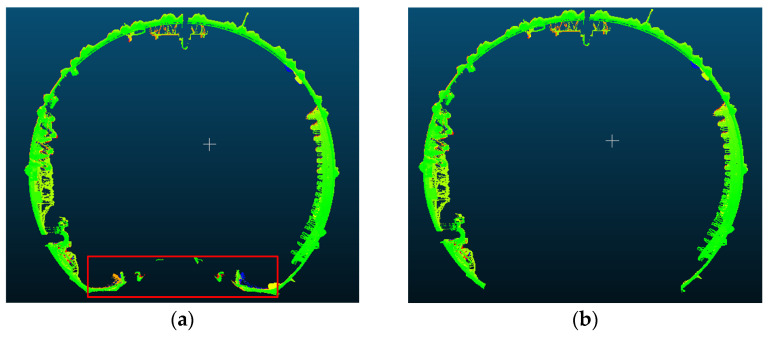
Separation of the tunnel and track bed: (**a**) before separating the tunnel from the track bed; (**b**) after separating the tunnel from the track bed.

**Figure 12 sensors-25-00356-f012:**
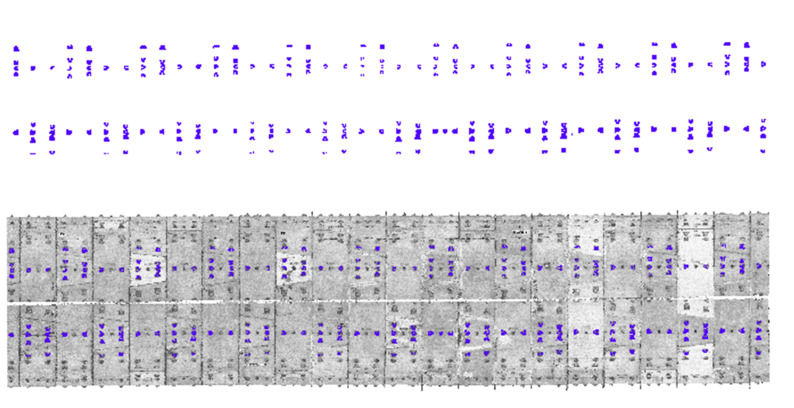
Extraction and cluster effect of the bolt holes at the top of the tunnel.

**Figure 13 sensors-25-00356-f013:**
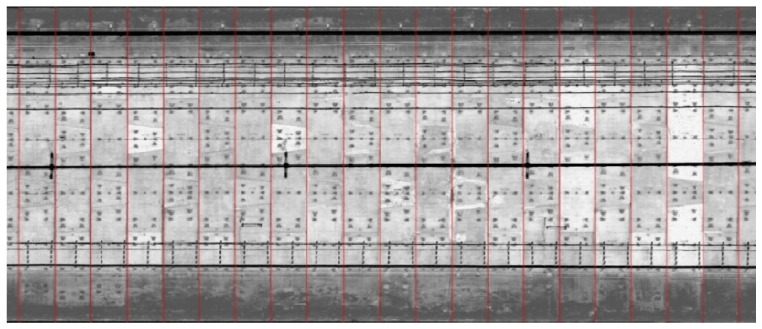
Tunnel mileage marking effect.

**Figure 14 sensors-25-00356-f014:**
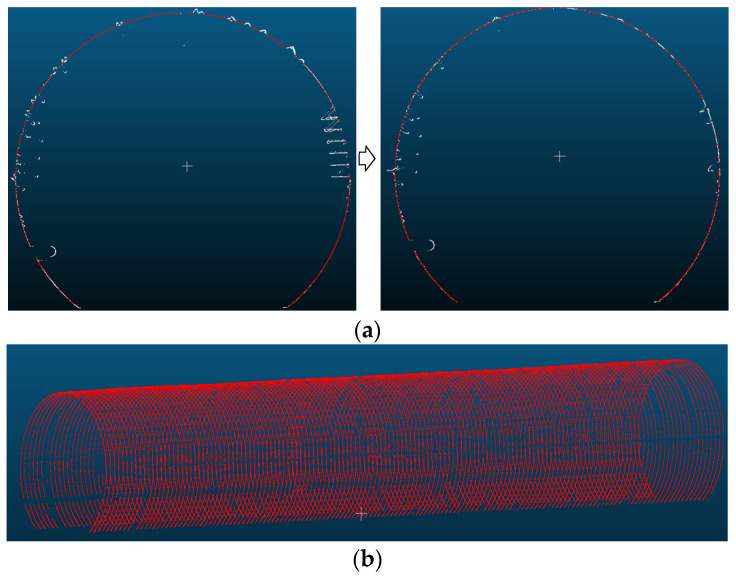
Section denoising and extraction: (**a**) section denoising; (**b**) section extraction.

**Figure 15 sensors-25-00356-f015:**
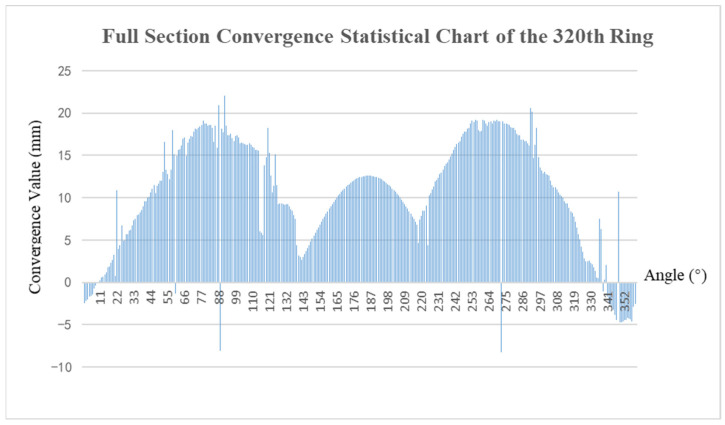
Full section convergence calculation results (320th ring).

**Figure 16 sensors-25-00356-f016:**
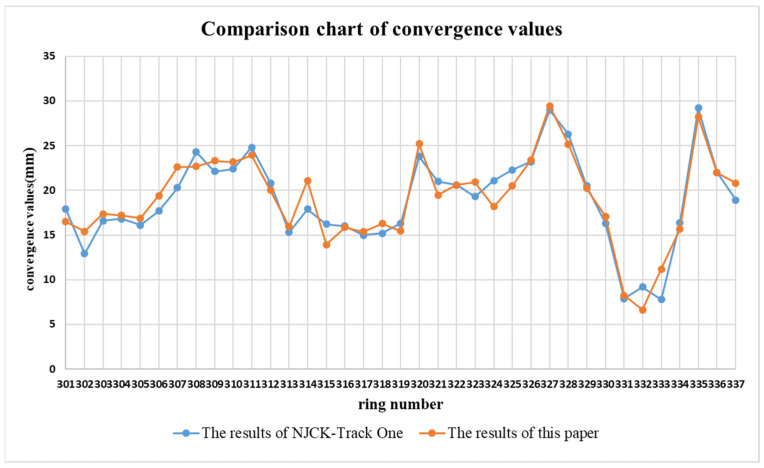
Comparison of convergence value calculation results.

**Figure 17 sensors-25-00356-f017:**
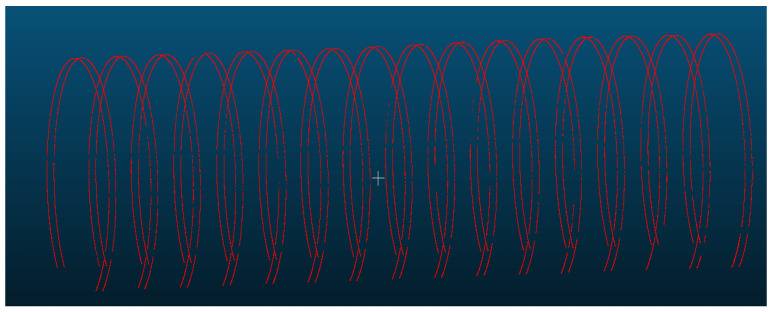
Cross-section extraction of inter-ring misalignment comparison.

**Figure 18 sensors-25-00356-f018:**
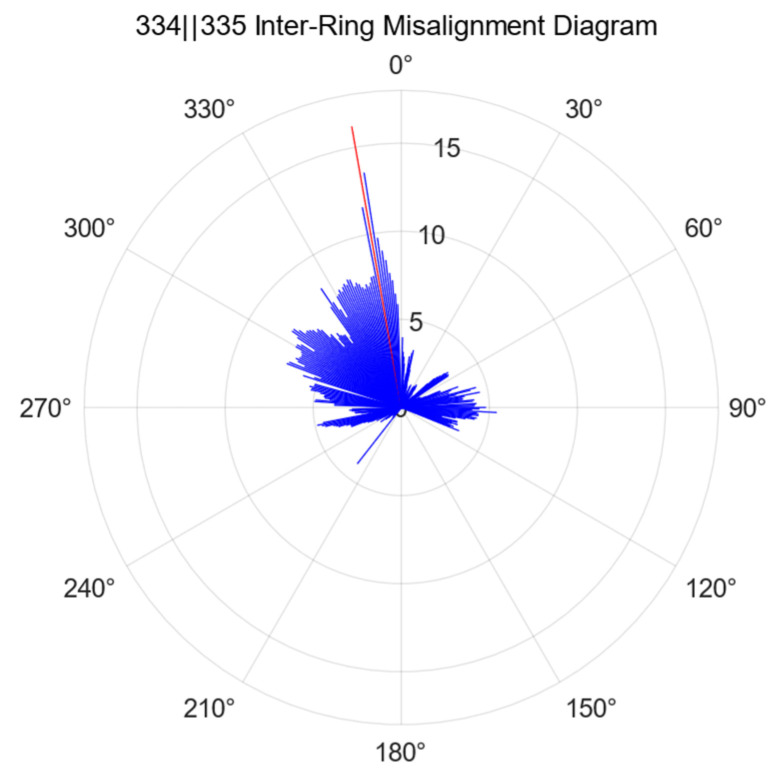
Schematic diagram of the inter-ring misalignment calculations.

**Figure 19 sensors-25-00356-f019:**
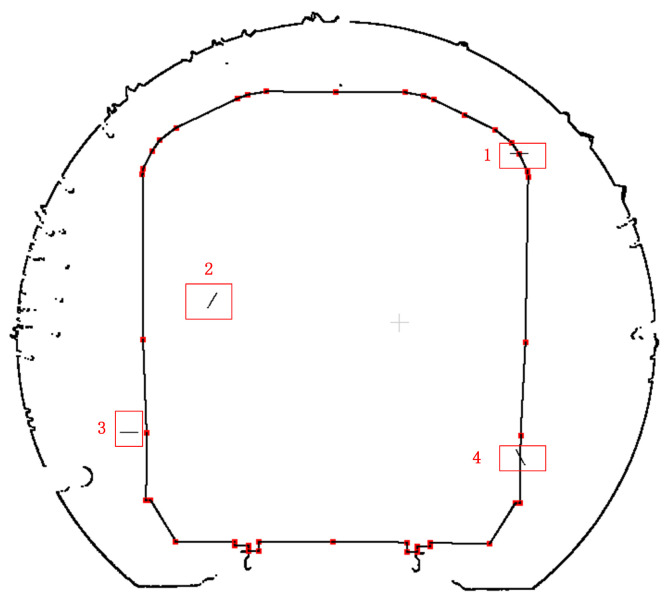
Schematic diagram of the simulated intrusion limit points.

**Figure 20 sensors-25-00356-f020:**
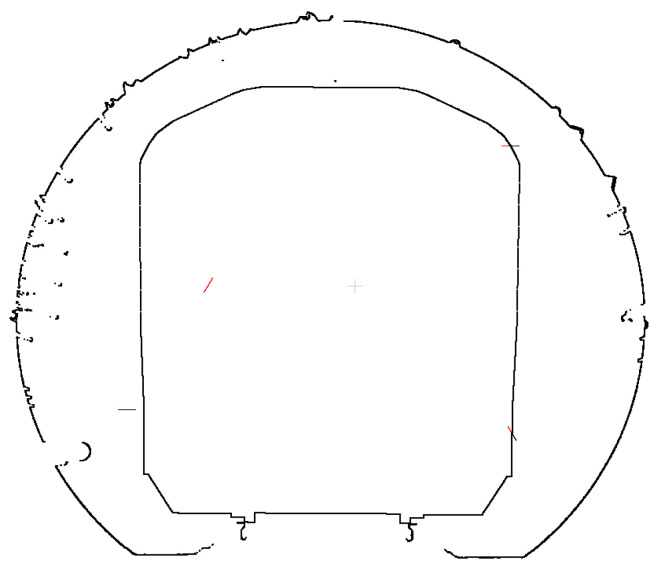
Results of limit point detection.

**Table 1 sensors-25-00356-t001:** Main technical parameters of ACT926T-10 dual-axis tilt sensor.

Technical Parameters	Specifications
Measurement range	±10°
Absolute accuracy (−40–80 °C)	0.001°
Resolution	0.0005°
Maximum measurement frequency	100 Hz
Average working time	≥55,000 h/time
Measurement axis	X, Y axis
Communication interface	RS232

**Table 2 sensors-25-00356-t002:** Ellipse fitting results of the partial section.

Ring Number	X-Coordinate of Center Point (m)	Z-Coordinate of Center Point (m)	Long Axis (m)	Short Axis (m)	Deflection Angle (°)
301	−0.0047	1.1469	5.5.650	5.5040	2.347
302	0.0019	1.1489	5.5154	5.5036	171.358
303	0.0129	1.1482	5.5174	5.5030	178.832
304	0.0210	1.1496	5.5172	5.5042	0.013
305	0.0303	1.1454	5.5169	5.5040	6.090
306	0.0356	1.1429	5.5194	5.5016	0.675
307	0.0386	1.1366	5.5226	5.4971	8.309
308	0.0388	1.1314	5.5227	5.4994	4.883
309	0.0379	1.1259	5.5233	5.4966	6.599
310	0.0356	1.1192	5.5232	5.5052	5.466

**Table 3 sensors-25-00356-t003:** Convergence calculation results of the partial section.

Ring Number	Convergence Value (mm)	Ellipticity	Ring Number	Convergence Value (mm)	Ellipticity
301	16.5	2.27	311	23.9	4.37
302	15.4	2.16	312	20.0	3.80
303	17.4	2.61	313	15.9	0.85
304	17.2	2.37	314	21.1	2.83
305	16.9	2.35	315	13.9	0.83
306	19.4	3.24	316	15.9	2.29
307	22.6	4.63	317	15.4	0.93
308	22.7	4.23	318	16.3	2.44
309	23.3	4.86	319	15.4	1.34
310	23.2	3.26	320	25.2	5.13

**Table 4 sensors-25-00356-t004:** Ring-to-ring misalignment calculation results of the partial section.

Ring Number	Maximum Misalignment (mm)	Maximum Misalignment Position (°)
322||323	7.38	349
323||324	9.54	280
324||325	4.35	332
325||326	11.98	292
326||327	8.48	325
327||328	8.25	10
328||329	10.72	350
329||330	8.87	293

**Table 5 sensors-25-00356-t005:** Intrusion detection results table.

Mileage Value of Detected Section (m)	Angle of Encroachment (°)	Encroachment Limit Length (mm)
24.32	41.98–43.38	74.7
24.32	120.61–121.73	69.3
24.32	284.77–292.12	142

## Data Availability

The data that support the findings of this study are available from the corresponding author upon reasonable request.

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
