# Peer review of "Development of a Mobile Laser Measurement System for Subway Tunnel Deformation Detection"

_sensors, 2025, doi:10.3390/s25020356_

Round 1

Reviewer 1 Report

Comments and Suggestions for Authors

1. The manuscript is detailed but overly dense in several sections, particularly in the Methods and Results. Simplifying the descriptions with concise summaries or visual aids (e.g., flow diagrams for the synchronization process or graphical representations of convergence results) would enhance readability.

2. The manuscript lacks a benchmarking comparison with existing systems like Leica Sitrack-one. Providing a quantitative evaluation of accuracy against these systems would strengthen the claims of novelty.

3. The experimental validation conducted on the Metro is commendable. However, a comparison of the system's performance with ground-truth measurements is missing. 

Reviewer 2 Report

Comments and Suggestions for Authors

In this manuscript, the development of a Mobile Laser Measurement System for Subway Tunnel Deformation Detection is presented. The measured results are of interest to engineering practice. However, there are several issues that need further clarification and explanation.

1.     On P103, Figure 1, two positions for installing GNSS antennas are designed above the vehicle body. Please consider whether to install GNSS and suggest the purpose of installation and the basis for selecting these positions.

2.     On P165, please provide the device model and related parameters such as accuracy for the 2D inclination sensor.

3.     On P219, a statistical filtering algorithm is initially used to eliminate outlier noise points. Please provide the name of the traditional filtering algorithm.

4.     On P229, to improve algorithm efficiency, the extraction of the central axis is converted from three-dimensional space to two-dimensional space. During the experiment, was there an attempt to extract the central axis in three-dimensional space? After converting to two-dimensional space, what is the error compared to the three-dimensional space? Please provide an explanation.

5.     On P293, Figure 6 is blurry; a clearer image is suggested.

6.     On P327, Figure 7 is blurry; a clearer image is suggested.

7.     On P450, the manuscript uses the ray method to detect the locomotive's ultimate intrusion. Compared to other methods, this approach has simpler judgment rules and lower computational complexity. However, its accuracy needs further clarification and explanation.

8.     On P492, for tunnel point cloud preprocessing and mileage markers, statistical filtering is influenced by the number of neighboring points queried in distance statistics and the threshold value. In this experiment, the number of neighboring points attempted is set to 70, and the threshold is defined as twice the standard deviation. Please provide the basis for these selections and explain their impact on the analysis results. Additionally, approximately 1,500 noise points were removed; it is suggested to clarify whether any points were incorrectly deleted.

9.     On P505, after determining the central axis, the shield tunnel is separated and dismantled based on the empirical value of the shield tunnel angle. It is suggested to provide the empirical value of the shield tunnel angle.

10.  On P518, using the clustering results, calculate the coordinate range of similar bolt holes, and determine the mileage values of the ring center and ring seam positions within the tunnel. It is recommended to compare and analyze the actual bolt holes with the calculated ones.

11.  On P539, in Table 1, why are the coordinates of 301 negative? Why are the deflection angles of 302 and 303 relatively larger compared to other positions? A detailed explanation is suggested.

12.  On P583, the Maximum misalignment position (°) for ring numbers 327328 is smaller compared to other positions. It is suggested to provide additional explanations. Additionally, the maximum misalignment between rings 334 and 335 occurs at 350°; it is recommended to supplement the reasons for this.

Round 2

Reviewer 1 Report

Comments and Suggestions for Authors

N/A